# Nutritional Composition, Bioactive Properties, and Sensory Evaluation of Breadsticks Enriched with Carp Meat (*Cyprinus carpio*, L.)

**DOI:** 10.3390/foods14234066

**Published:** 2025-11-27

**Authors:** Grzegorz Tokarczyk, Grzegorz Bienkiewicz, Katarzyna Felisiak, Patrycja Biernacka, Tomasz Krzywiński, Marek Bury, Cezary Podsiadło, Eire López Arroyos

**Affiliations:** 1Department of Fish, Plant and Gastronomy Technology, Faculty of Food Science and Fisheries, West Pomeranian University of Technology in Szczecin, 70-310 Szczecin, Poland; grzegorz.tokarczyk@zut.edu.pl (G.T.); katarzyna.felisiak@zut.edu.pl (K.F.); krzywy-t@o2.pl (T.K.); 2Department of Commodity Science, Quality Assessment, Process Engineering and Human Nutrition, Faculty of Food Science and Fisheries, West Pomeranian University of Technology in Szczecin, 70-310 Szczecin, Poland; grzegorz.bienkiewicz@zut.edu.pl; 3Department of Agroengineering, West Pomeranian University of Technology in Szczecin, 70-310 Szczecin, Poland; marek.bury@zut.edu.pl (M.B.); cezary.podsiadlo@zut.edu.pl (C.P.); 4Department of Food Science and Technology, University of Zaragoza, C. de Pedro Cerbuna Street 12, 50009 Zaragoza, Spain

**Keywords:** *Cyprinus carpio*, bioactive compounds, amino acid profile, available lysine, fatty acid, functional foods

## Abstract

Cereal-based snacks, such as breadsticks and salty sticks, are widely consumed but nutritionally poor, lacking protein, essential amino acids, bioactive compounds, and functional lipids. Enhancing these products with fish-derived ingredients could provide a novel approach to improving their nutritional and functional value. This study investigated the effect of incorporating carp meat (0–30%) into breadsticks in terms of their composition, amino acid and fatty acid profiles, mineral content, antioxidant activity, lipid stability, and sensory attributes. Fortification with carp meat substantially improved nutritional value, with a significant increase in essential amino acids, especially available lysine, and long-chain ω3 fatty acids. Antioxidant activity increased depending on the amount of fish meat added, contributing to reduced total oxidation values. Sensory evaluation revealed that 10–15% fish addition is the optimal range, combining improved nutritional quality with high consumer acceptability. The results show that carp-enriched breadsticks represent a promising functional food concept that can deliver bioactive nutrients in a familiar snack form. This approach highlights the feasibility of fortifying food with fish as a strategy to diversify healthy snacks, increase fish consumption, and provide new opportunities for innovation in the food industry.

## 1. Introduction

Salty, baked snacks like breadsticks, salty sticks, and pretzels are among the most popular convenience foods in the world. Their popularity stems from their convenience, affordability, and sensory qualities [1]. However, such products are typically high in energy and low in nutrients. They are characterized by high levels of refined carbohydrates and salt, while being low in high-quality proteins, essential fatty acids, and bioactive compounds [2]. With growing consumer demand for healthier alternatives, reformulation strategies to improve the nutritional profile of snacks have become a major focus of science and the food industry. One promising approach is to fortify cereal products with animal or seafood proteins, which can significantly improve their amino acid balance, fatty acid profile, and functional properties [3,4,5,6,7]. Fish protein hydrolysates have been added to pasta, bread, and bakery products, increasing the protein levels and antioxidant qualities while maintaining sensory attributes [8]. Similarly, fish flour and ground fish from different species have been utilized to enhance pasta, crackers, and baked goods, leading to improved protein quality and a better balance of essential amino acids [9,10]. Seaweeds have been utilized as functional components in bakery products and various starchy foods, enhancing mineral levels, fiber, and bioactive substances while altering textural characteristics [11,12]. Moreover, microencapsulated fish oils have been effectively incorporated into bread and biscuits to increase their ω3 fatty acid levels, showing that widely consumed cereal-based products can act as suitable vehicles for health-benefiting long-chain PUFA when suitable formulation techniques are utilized [12,13]. Although direct consumption of fish is recommended, many consumers, especially younger ones, prefer convenient snacks over traditional fish products. Therefore, incorporating fish ingredients into commonly consumed snacks may be a promising strategy for increasing bioactive nutrient intake. Limited research exists on directly incorporating fish meat into baked snack sticks, which could simultaneously provide nutritional enrichment, bioactive functionality, and acceptable sensory quality.

Common carp (*Cyprinus carpio* L.) is one of the most widely farmed freshwater fish species in Europe and Asia, accounting for a significant share of aquaculture production [14]. As a species highly suited to a variety of environmental conditions, carp is cultivated in a manner that is both efficient and sustainable, providing a readily available and economically significant source of animal protein. Fish and seafood are a valuable source of essential amino acids, long-chain polyunsaturated fatty acids (especially omega-3 fatty acids), minerals, and antioxidant compounds [15]. Nutritionally, carp meat is characterized by a high-quality protein fraction containing all essential amino acids, including lysine, which is typically limiting in cereal-based products [16]. Consuming fish products regularly has been well established to provide a variety of health advantages. These benefits encompass heart protection, with an influence on lipid metabolism and a reduction in inflammation; support for cognitive growth and brain protection, attributed to omega-3 fatty acids; and general anti-inflammatory and immune-modulating effects, linked to bioactive substances found in fish [17,18]. Considering these characteristics, carp serves as a significant raw material for creating fortified or functional foods designed to enhance the dietary quality of groups with limited fish consumption.

The aim of the study was to develop and evaluate snacks enriched with various amounts of carp (*Cyprinus carpio*) meat to increase their nutritional value and functional properties. Unlike previous studies, which have primarily focused on enriching baked goods with plant-based ingredients or directly processing fish products, our study introduces a novel concept: combining a popular cereal-based snack with aquatic ingredients. This approach offers a dual benefit: it transforms a nutritionally poor but commonly consumed product into a source of high-quality proteins, essential amino acids (including digestible lysine), polyunsaturated fatty acids, minerals, and antioxidant compounds, and also assesses consumer acceptance. By integrating comprehensive analyses of chemical composition, lipid stability, antioxidant activity, sensory characteristics, and texture parameters, this study provides a holistic assessment of fish-enriched snacks as an innovative approach to delivering bioactive fish ingredients to a broader range of consumers.

## 2. Materials and Methods

### 2.1. Sample Preparation

The following ingredients were used to produce the breadsticks snacks: type 450 wheat flour, table salt, dried yeast, sugar, rapeseed oil, baking powder, and drinking water. All ingredients were purchased at a local Intermarché store in Szczecin, Poland.

Manageable remains of industrial fileting of farmed carp (*Cyprinus carpio*), such as backbones, collarbones, or parts of fins, were obtained fresh from the fish processing plant in Poland and transported in ice to the laboratory. The fish remains were washed with chilled potable tap water and then passed through a drum separator type NF 13DX, having 4 mm diameter holes (Bibun, Fukuyama, Japan). They were then cleaned thoroughly in a screw separator type SUM 420, with 2.5 mm diameter holes (Bibun, Japan). That was how clean comminuted fish meat was received.

The dough was prepared by mixing flour with dissolved yeast, salt, sugar, and baking powder, and adding oil and water until a uniform consistency was achieved. A measured amount of carp meat was added to the dough (Table 1). All ingredients were mixed, and after kneading, the dough was proofed at 30 °C for 30 min with 75% of relative humidity in a fermentation chamber (Grafen, Nano Bakery 8xGN 1/1, Robakowo, Poland). The dough was then extruded in a P3 pasta extruder (La Monferrina, Asti, Italy) with a modified extrusion die, and the formed snacks were baked in an electric oven (Hendi H90S, Robakowo, Poland) at 220 °C for 7 ± 2 min, until golden brown. After baking, the products were cooled at room temperature (21 ± 2 °C) for 1 h, then packaged in plastic boxes with lids and stored at room temperature until analysis (no longer than 24 h).

### 2.2. Basic Composition

In accordance with the recommendations of the Association of Official Analytical Chemists (AOAC), the subsequent parameters were examined: dry matter, total nitrogen (analyzed via the Kjeldahl method using the Kjeltec KT 200 apparatus from Labtec Line, FOSS, Warszawa, Poland), lipid (fat) content (determined through the Soxhlet method), and ash content (assessed using the dry mineralization technique) [19]. Nitrogen-free extract (NFE), representing the carbohydrate fraction, was calculated by difference as recommended in proximate analysis standards [20]:(1)NFE% = DM% − (LC% + PC% + AC% + FC%) where DM%—dry matter (%);LC%—lipid content (%);PC%—protein content (%);AC%—ash content (%);FC%—fiber content (%).

Total dietary fiber was determined by the enzymatic-gravimetric AOAC method 991.43 [21]. Samples were homogenized, digested sequentially with α-amylase, protease, and amyloglucosidase, then filtered, washed, dried, and ashed. Fiber content was calculated from mass differences after correcting for ash and protein residues (Kjeldahl). Each sample was analyzed in triplicate to ensure reliability.

### 2.3. Content of Micro and Macro Elements

The analysis of selected elements (P, K, Mg, Na, Fe, Ca, Ni, Cr, Pb, Cd, Mn, Co, Cu, Zn) was performed with an atomic absorption spectrometer (Thermo Fisher Scientific iCE 3000 Series, Waltham, MA, USA). The samples were mineralized and then tested in various dilutions [22].

### 2.4. Antioxidant Activity

Antioxidant properties were assessed using three methods: Trolox equivalent antioxidant capacity (TEAC) [23], DPPH radical scavenging activity [24], and ferric reducing antioxidant power (FRAP) [25]. Results were expressed as Trolox equivalents (µmolTE) per g of sample.

Total phenolic content (TPC) was measured in methanolic extracts using the Folin–Ciocalteu method described by Turkmen et al. [26]. Briefly, diluted extract was mixed with Folin–Ciocalteu reagent and sodium carbonate, incubated in the dark for 2 h, and absorbance was read at 750 nm with a spectrophotometer (Thermo Spectronic, Horsham, UK). Results were expressed as mg gallic acid equivalents per g (mg GAE/g).

### 2.5. Fatty Acid Profile

Fatty acid methyl esters (FAMEs) were prepared from lipid extracts by alkaline hydrolysis with 0.5 N sodium methylate [27,28]. FAMEs were analyzed by gas chromatography–mass spectrometry (GC–MS, Agilent 7890A, Santa Clara, CA, USA) equipped with a split/splitless injector. Separation was achieved on an SP™ 2560 capillary column (100 m × 0.25 mm i.d., 0.20 μm film; catalog no. 24056) under the following conditions: split ratio 1:50, injector at 220 °C, detector at 240 °C, and oven program from 140 °C (5 min hold) to 240 °C at 4 °C/min, with a total runtime of 45 min [29]. Nonadecanoic acid (C19:0; CAS 646-30-0, Merck, Warsaw, Poland) was used as the internal standard.

### 2.6. Lipid Value

The peroxide value (PV) was determined in the lipids extracted using the Bligh and Dyer method [27], according to EN-ISO 3960:2017 [30], based on the iodometric determination of iodine liberated by the peroxides with a starch indicator and a sodium thiosulfate standard solution. Results were expressed as milliequivalents of active oxygen per kilogram of lipids (meqO_2_/kg of fat). Anisidine value (AsV) was determined in the lipid extract according to the EN-ISO 6885:2007 method [31], based on the reaction between α- and β-unsaturated aldehydes and p-anisidine reagent. AsV was expressed as 100 times the absorbance measured at 350 nm (Thermo Scientific, Genesys 20, Waltham, MA, USA) in a 1 cm path length cuvette from a solution containing 10 mg of lipid in 1 mL of reaction medium. The acid value (AV) was determined by the titration of 0.1 N KOH in methanol, according to EN ISO 660:2020 [32]. The results were presented in milligrams of KOH per gram of fat.

The TOTOX index, which reflects overall lipid oxidation by combining primary (PV) and secondary (AsV) oxidation products, was then calculated. Total oxidation (TOTOX) products were calculated as follows:(2)TOTOX = 2PV + AsV

### 2.7. Amino Acid Profile

Amino acid profiles in samples were examined following acid hydrolysis in 6 N HCl for 22 h at 110 °C in glass tubes filled with nitrogen. Cystine and methionine were assessed as cysteic acid and methionine sulfone, respectively, through oxidation using performic acid before their digestion with 6 N HCl [33,34]. Tryptophan was measured using the method described by Landry, Delhaye, and Jones (1992), following the alkaline hydrolysis of each sample [35]. Chromatographic analysis was conducted employing an HPLC—Agilent 1260 Infinity II liquid chromatograph (Agilent Technologies, Inc., Santa Clara, CA, USA) with a PDA detector. Each sample was analyzed in triplicate, and quantification was carried out using external standards.

### 2.8. Available Lysine

Available lysine was determined using the 1-Fluoro-2,4-dinitrobenzene (FDNB) procedure of Carpenter (1960) [36] with the modification proposed by Both (1971) [37].

### 2.9. Texture Parameter Analysis

The textural attributes of the snacks were analyzed for hardness and crispness according to the method described by Levent and Bilgiçli (2013) [38]. A TA. XT plus C^®^ Texture Analyzer (Stable Micro Systems^®^, Godalming, UK) with a 3-point bend rig (HDP/3PB) connected to a 25 kg load cell was used, with the following parameters: test speed of 1.0 mm s^−1^, posttest speed: 1.5 mm s^−1^, trigger force of 0.005 N, and data acquisition rate of 200 points s^−1^. Ten snacks from each batch were tested. Hardness was measured as the peak force, representing the maximum resistance of each snack against a blade, and occurred when the sample began to break. Crispness was defined as the distance to peak force, and snacks with greater distance values were less fracturable.

### 2.10. Sensory Evaluation

Sensory evaluation was performed 24 h after baking by 10 trained in the sensory laboratory of the Faculty of Food Sciences and Fisheries, West Pomeranian University of Technology in Szczecin. The sensory analysis performed in the study did not pose a threat to the physical or mental well-being of the participants, did not pose a risk of privacy violations, and excluded the possibility of other social or legal repercussions related to compliance with ethical guidelines and standards. All participants were adults, and they were not intellectually disabled. Therefore, the permission of the Committee on Ethics in Human Beings Research of West Pomeranian University of Technology in Szczecin was not required. The individuals participating in the product evaluation gave their informed and voluntary consent, and had been trained in the sensory sensitivity standards PN-ISO 5496:1997 and PN-ISO 3972:2016-07 [39,40]. The organoleptic evaluation was performed under conditions compliant with BS EN ISO 8589:2010+A1:2014 [41]. In these prepared samples, the following were assessed: appearance, color, aroma, flavor, and texture using a 5-point scale (ranking from 1—poor to 5—very good). Additionally, overall acceptability was assessed using a 9-point scale (ranking from 1—very bad to 9—very good). Each samples were assessed individually by the participant on prepared cards in two independent sessions. Additionally, the intensity of flavor and aroma perception in snacks assessed using a 9-point hedonic scale (0—imperceptible, 1—very weak, 2—weak, 3—rather weak, 4—moderately weak, 5—moderate, 6—fairly strong, 7—strong, 8—very strong, 9—extremely strong) according to a sheet developed for each of the two sessions in accordance with Baryłko-Pikielna and Matuszewska [42], based on which mean values, standard deviation were calculated from the obtained data and sensory profiles were determined. To ensure inter-panelist reliability, agreement among the ten trained assessors was evaluated with Kendall’s tau (τ) rank correlation. This method is appropriate for the ordinal nature of the sensory ranking data and allows assessment of the strength and direction of association between evaluators. For all descriptive attributes (appearance, color, aroma, flavor, texture, and overall acceptability), Kendall’s tau values were τ = 1.00, indicating perfect agreement and demonstrating that panelists applied the ranking scales consistently and reproducibly across both evaluation sessions.

### 2.11. Statistical Analysis

Analyses were performed in triplicate. Results were statistically analyzed using one-way analysis of variance (ANOVA) in StatSoft Statistica 13.3 (Statsoft, Tulsa, OK, USA). The ANOVA *p*-value was set at 0.05, and significant differences between samples were tested using Tukey’s post hoc test (*p* < 0.05). Agreement among the ten trained assessors was evaluated with Kendall’s tau (τ) rank correlation [43].

## 3. Results and Discussion

### 3.1. Basic Composition

The carp meat used in the formulation is a lean, protein-dense raw material with moisture typical of freshwater fish [44], making it well-suited for enhancing the nutritional value of cereal-based snacks (Table 2). Enriched breadsticks significantly influence their basic chemical composition. The dry matter content ranges from 94.24 ± 0.25% in the control sample (S) to 93.51 ± 0.12% in the sample with 30% fish (S30). The fat content of these snacks increases proportionally with the increase in fish content, ranging from 18.02% in the control sample to 23.16 ± 0.16% in S30. Fish lipids are known to provide beneficial long-chain polyunsaturated fatty acids, particularly omega-3 fatty acids, which are associated with cardioprotective and anti-inflammatory effects [18]. Therefore, despite the increase in total fat content, the qualitative fatty acid profile of breadstick snacks may provide a nutritional advantage compared to conventional formulas. An increase is also observed for protein, which increases significantly from 9.73 ± 0.16% in the control sample to 19.52 ± 0.11% in S30. Ash content gradually decreases from 4.18 ± 0.04% in S30 to 2.21 ± 0.02% in the sample with the highest carp meat addition. Cereal flours typically contain more mineral residues after combustion compared to fish meat [45]. NFE values decrease steadily as the amount of carp meat increases, showing that carbohydrate-rich flour is being substituted with fish meat that is higher in protein and fat. This change demonstrates a significant decrease in carbohydrate density and the energy contribution from starch as the carp content rises. However, detailed elemental analysis reveals that enriching these snacks with carp meat selectively modifies the mineral composition, rather than uniformly (Table 3). Enriching flour products with protein-rich fish meat reduces the total fiber content of the diet, as these ingredients are inherently fiber-free, resulting in a dilution effect in the final product composition.

Macrominerals of the carp meat (Table 3) align closely with established profiles for *Cyprinus carpio*, which generally show potassium and phosphorus as the primary macrominerals, together with moderate amounts of magnesium, calcium, and sodium typical of freshwater fish muscle [44,46]. Nutritionally significant levels of iron, zinc, and manganese are present, consistent with earlier findings on the mineral content of carp muscle tissue [47]. In the case of breadsticks, phosphorus content decreases progressively with increasing carp addition, falling by approximately 25% from S0 to S30. A similar downward trend is observed for calcium and sodium, which declined by about 2% at the highest level of fish fortification. Sodium content remained relatively stable across all samples, reflecting the even addition of salt during formulation rather than a fish-specific effect. This trend can be explained by the dilution of phytate phosphorus from cereals, which is abundant in flour, by rice proteins, but also by heat treatment [48]. Potassium and magnesium concentrations systematically increased with the addition of carp meat. Potassium increases by approximately 70% in S30 compared with the control, and magnesium increases by roughly 26%. These results are consistent with the established role of fish muscle as a rich reservoir of intracellular electrolytes, which are retained within the muscle cells [49]. Iron concentration increases moderately, by around 6% in S30 relative to S0. While this may seem a small increase, its nutritional significance is substantial. Fish provides highly bioavailable heme iron, which can more effectively replenish iron deficiencies than plant sources [50]. An increase in zinc content is also observed (abou 13%), which is beneficial given its importance for immune and enzymatic functions [51]. The manganese and chromium content remain low and do not show significant fluctuations. Importantly, toxic heavy metals such as cadmium, lead, cobalt, and copper are undetectable in all samples, confirming the safety of carp meat as a snack ingredient. This finding is crucial, as it confirms the safety of using carp meat in this context, alleviating concerns about the potential bioaccumulation of environmental contaminants in aquatic organisms [52,53]. This ensures that the nutritional benefits of fortification are not offset by toxicological risks.

### 3.2. Antioxidant Activity

The addition of carp meat may enhance the antioxidant potential of the product. In our study, it shows a significant ability to act as an antioxidant, with moderate values for TEAC and FRAP indicating its capacity to neutralize free radicals and decrease oxidants (Table 4). Similarly, the DPPH activity demonstrated a noticeable ability to scavenge radicals. The total phenolic content is found to be low, suggesting that the antioxidant properties of carp meat mainly come from non-phenolic components characteristic of fish muscle. Carp meat has been shown to contain a high proportion of aromatic and hydrophobic amino acids, notably tyrosine, which is well known for its antioxidant capability. The antioxidant activity of breadsticks is significantly influenced by the carp meat content. In particular, in research, Wong et al. (2025) report that low-molecular-weight peptide fractions (dominantly dipeptides and free amino acids < 2.4 kDa) generated by simulated gastrointestinal digestion contain a high fraction of tyrosine and exhibit strong radical-scavenging and iron-chelating activities [54]. TEAC (trolox equivalent antioxidant capacity) values increase systematically with enrichment, from 1.71 ± 0.18 μmol TE/g in the control sample to 3.50 ± 0.26 μmol TE/g in S30. A similar trend is observed for FRAP (ferric reducing antioxidant power), which increases from 68.7 ± 1.2 μmol TE/g in the control sample to 94.7 ± 2.2 μmol TE/g in S30, indicating a dose-dependent increase in reducing power. DPPH radical scavenging activity also improves gradually, with values increasing from 0.516 ± 0.031 μmol TE/g in the control sample to 0.738 ± 0.021 μmol TE/g in S30. These results indicate that the addition of carp meat improves the antioxidant potential, but this improvement is not directly related to phenolic compounds. Instead, the increased activity may reflect the contribution of amino acids and other antioxidant components derived from fish muscle proteins and lipids [55,56]. Fish proteins have been reported to generate peptides with potent radical scavenging and reduction properties during processing or digestion [57]. Although heat treatment may promote limited peptide liberation [58], actual extent of peptide formation and their survival during storage or digestion is unknown. Additionally, antioxidant peptides in fish muscle are usually generated through regulated enzymatic hydrolysis [59], with their natural development during baking being inconsistent and depending on the product. This variability also affects considerations regarding shelf life: while certain peptides or amino acids might aid in radical scavenging, the simultaneous rise in polyunsaturated fatty acids may increase the likelihood of lipid peroxidation occurring more rapidly over time [60]. Therefore, even though there is a definitive dose-dependent enhancement in chemical antioxidant measures, more investigation is needed, including peptide profiling, digestion assessments, and tests on storage stability.

Total polyphenol content (TPC) does not follow the same linear pattern. The control sample contains 0.678 ± 0.009 mg GAE/g, while values varied slightly with the proportion of carp meat, ranging from 0.612 ± 0.010 to 0.673 ± 0.028 mg GAE/g, without a clear upward trend. The lack of a significant increase in TPC suggests that phenolic compounds from cereal products dominated this fraction, while carp meat provides mainly non-phenolic antioxidant components. This pattern is consistent with previous studies that have shown that protein hydrolysates and bioactive peptides derived from seafood may exert significant antioxidant activity, regardless of phenolic content [61].

### 3.3. Fatty Acids Profile and Lipid Values

The fatty acid profile of carp meat (Table 5) is dominated by monounsaturated fatty acids (MUFA), which constitute the largest proportion of total fat, primarily due to the high contribution of oleic acid (C18:1 ω9), a characteristic feature of freshwater cyprinid species [62]. Saturated fatty acids (SFA) form the second major group, with palmitic acid (C16:0) as the principal component. Polyunsaturated fatty acids (PUFA) remain substantial, including both ω6 and ω3 families, with linoleic acid (C18:2 ω6) and α-linolenic acid (C18:3 ω3) as notable contributors.

The addition of carp meat to the breadsticks significantly changes the fatty acid (FA) profile. SFA constitutes the largest group in the total profile and shows a gradual increase with increasing amounts of added fish. However, the content of MUFA remains relatively stable, ranging from 58.50 ± 1.12% to 60.19 ± 0.67%. PUFA constituted the second-highest total fatty acid content across all groups, with a small but consistent increase in enriched samples compared to the control sample. The slight increase in SFA content observed after the addition of carp meat can be attributed to palmitic acid. While elevated SFA intake is often a concern in modern diets, the context here is crucial. The increase is not occurring in isolation but is part of a broader, more beneficial lipid restructuring. Maintaining high levels of MUFAs, dominated by oleic acid, further confirms the beneficial lipid profile of the enriched products, as oleic acid is known to exert hypocholesterolemic and anti-inflammatory effects [63,64].

The oleic acid content is highest in S30 (57.32 ± 0.21%), while the linoleic acid content decreases slightly from 20.52 ± 0.15% (S) to 19.86 ± 0.16% (S30). Oleic acid is well-documented for its hypocholesterolemic effects, specifically its ability to reduce low-density lipoprotein (LDL) cholesterol without adversely affecting high-density lipoprotein (HDL) levels [65]. Particularly significant is the proportion of long-chain polyunsaturated fatty acids ω3 (LC ω3 PUFA), including α-linolenic acid (C18:3 ω3), eicosapentaenoic acid (EPA, C20:5 ω3), and docosahexaenoic acid (DHA, C22:6 ω3). These are absent in the control sample, but increase with the addition of carp meat. These changes in the fatty acid profile have significant nutritional significance. This shift is of paramount importance from a public health perspective. Modern Western Diets are often characterized by an excessively high omega-6 to omega-3 ratio, which is pro-inflammatory and linked to an increased risk of cardiovascular diseases, cancer, and inflammatory disorders [66]. Increased intake of long-chain polyunsaturated fatty acids ω3, especially EPA and DHA, is associated with cardioprotective, neuroprotective, and anti-inflammatory benefits [67,68]. Although their absolute contribution to the total fatty acid content is small, the presence of these compounds in breadsticks represents a significant innovation, as these products typically do not contain polyunsaturated fatty acids.

The carp meat shows low levels of lipid value, as reflected by the peroxide value (PV) of 1.81 ± 0.12 meq O_2_/kg fat, indicating minimal formation of primary oxidation products. The anisidine value (AsV) is 3.22 ± 0.10, demonstrating only slight accumulation of secondary aldehydic compounds. The acid value (AV) is also low—0.52 ± 0.05 mg KOH/g fat, confirming limited hydrolytic lipid degradation. Consequently, the calculated TOTOX value of 8.32 ± 0.06 confirms that the overall oxidative status of the carp meat is suitable and characteristic of a fresh, non-oxidized raw material [69].

The acid value, which indicates the extent of hydrolysis and degradation of fats, increased gradually with the amount of added fish (Figure 1). The lowest value is recorded in the control sample (0.70 ± 0.06 mg KOH/g of fat), and the highest in S30 (1.98 ± 0.10 mg KOH/g of fat). This trend indicates greater liberation of free fatty acids in products enriched with fish remains, likely due to the higher proportion of polyunsaturated fatty acids, which are more susceptible to enzymatic and thermal hydrolysis during dough formation and baking [70]. Although the measured AVs are still within acceptable ranges for breadsticks, the increase indicates a potential greater susceptibility to lipid peroxidation during storage. Free fatty acids are known to enhance oxidative reactions, especially in PUFA-rich environments, as they lack the protective structure provided by esterified lipids and are more vulnerable to oxygen and pro-oxidant catalysts [60,71]. As a result, products with elevated fish content may have a reduced shelf life unless safeguarded by suitable packaging, antioxidants, or modified-atmosphere conditions [72]. This aspect is crucial for future product development and commercial uses, particularly in the realm of ω3-enriched foods, where oxidative stability serves as a vital quality indicator.

The peroxide value, an indicator of the main oxidation products (hydroperoxides), decreases with enrichment, from 5.63 ± 0.21 meq O_2_/kg of fat in the control sample to 2.18 ± 0.08 meq O_2_/kg of fat in S30 (Figure 2). Anisidine value, measuring secondary oxidation products (mainly aldehydes), also gradually increases with fish addition, from 2.22 ± 0.11 in S to 4.59 ± 0.28 in S30. This suggests that, despite the increase in antioxidant activity, some lipid secondary oxidation occurred. However, the simultaneous reduction in PV and TOTOX (Figure 3) indicates that the accumulation of secondary products is limited compared to the control sample. This observation confirms the protective effect of fish-derived antioxidants and highlights the role of nonphenolic compounds in stabilizing the lipid fraction of breadsticks [18].

TOTOX, which integrates peroxide and anisidine values to assess an overall measure of oxidative deterioration, shows a significant decrease from 13.48 ± 0.32 in S to 8.95 ± 0.29 in S30 (Figure 3). These results suggest that the inclusion of carp meat, despite increasing the content of oxidizable lipids, enhanced the oxidative stability of the products. This can be explained by the parallel increase in antioxidant activity observed in samples supplemented with fish. As shown by TEAC, FRAP, and DPPH assays, antioxidant potential consistently increases with the addition of carp meat. Fish proteins produce peptides with radical scavenging activity during processing [73], and endogenous antioxidants present in fish muscle (e.g., taurine, carnosine, and selenium compounds) may additionally inhibit lipid peroxidation [74,75,76]. These compounds compensate for the susceptibility to oxidation of higher PUFA content, resulting in lower PV and TOTOX values in breadsticks.

### 3.4. Amino Acid Profile and Loss of Available Lysine

The amino acid profile of the carp meat shows a composition typical of freshwater fish muscle, with glutamic acid, aspartic acid, leucine, lysine, and valine dominating the profile and contributing to its high nutritional value (Table 6). Essential amino acids (EAAs) represent a substantial proportion of total amino acids (TAA), confirming the high biological quality of carp proteins. The presence of hydroxyproline reflects collagen-rich connective tissue naturally occurring in fish meat [46,77]. Overall, the carp meat displays a well-balanced amino acid pattern characterized by high total amino acid content and a favorable essential amino acid (EAA) to non-essential amino acid (NEAA) ratio, consistent with previously reported profiles for carp meat [78].

Enriching breadsticks with carp meat significantly improves their nutritional value, particularly with respect to protein and lipid fractions. The significant increase in protein content is in line with expectations, as fish muscle is a rich source of high-quality proteins and essential amino acids [79]. This protein enrichment is important for cereal-based snacks, which are typically deficient in lysine and other essential amino acids [80]. Table 6 presents the amino acid profile of the snacks analyzed. TAA content increased systematically with the proportion of carp meat, representing a 44% increase compared to the control sample. Both EAA and NEAA content increase. According to Paoletti et al. (2024), indispensable amino acids should be considered individually because the limiting amino acid in a given food source defines its overall nutritional value [81]. Among the individual amino acids, glutamic acid and aspartic acid are the most abundant NEAAs. These amino acids contribute to the perceived umami Flavor and may improve sensory quality [82]. In terms of EAAs, leucine, lysine, and valine show the greatest increase. The nutritional standard for leucine for Healthy North American adults is 36 mg/kg/d [83]. The highest leucine content is found in sample S30 (980.0 ± 24.0 g/100 g). Consuming even a modest portion of the fish-enriched breadsticks would contribute meaningfully to the recommended leucine. Lysine, the limiting amino acid in cereals, increased significantly from 143.0 ± 18.0 mg/100 g in the control sample to 980 ± 36.0 mg/100 g in S30—a six-fold increase. This confirms that enriching fish meat significantly improves the biological value of proteins in these snacks. Increases in methionine and threonine content have also been reported in breadsticks. With methionine content reaching 172.0 ± 21.0 g/100 g in sample S30, breadsticks provide a meaningful contribution toward the dietary reference intake of 12.6 mg/kg/day for healthy adults [84]. These amino acids are often limited in plant proteins, and their enrichment promotes better protein utilization and balances nitrogen metabolism [85,86]. Taurine is not detected in any sample. The increase in histidine, reaching 820 ± 12.0 mg/100 g in S30, is significant because histidine residues are associated with antioxidant activity of peptides, particularly in radical scavenging and metal chelating [57].

The addition of carp meat to the snacks affects the available lysine content (Table 7). Before baking, available lysine increases steadily with higher levels of fish addition, from 113.0 mg/100 g in the control sample to 887.0 mg/100 g in the breadsticks with 30% carp meat. After baking, lysine content follows a similar trend, rising from 68.0 mg/100 g in the control to 734.0 mg/100 g in S30. Assessment of available lysine before and after baking confirmed that heat treatment led to measurable losses. In the control sample, available lysine is reduced by 40%. In contrast, samples enriched with carp meat show significantly lower relative losses: from 24.4 ± 3.0% in S5 to only 17.3 ± 2.0% in S30. These results demonstrate that a higher protein content and its associated influence limit lysine degradation during heat treatment. Lysine retention is of significant nutritional importance because lysine is the first limiting amino acid in cereal products and is particularly susceptible to loss in Maillard-type reactions with reducing sugars [87]. Data indicate that enrichment with carp meat not only increases the total lysine content but also improves its retention after heat treatment, ensuring that the nutritional value of the final product is maintained. Moreover, the lower percentage losses observed at higher carp additions can be partly attributed to greater product moisture: formulations that retain more water during baking show smaller declines in available lysine. Available lysine is lost primarily through Maillard reactions with reducing sugars. Reaction rates rise as products dry and temperatures increase; conversely, independent of higher moisture, exhibit slower browning kinetics, which protects ε-amino groups of lysine. This moisture–lysine relationship during baking is well documented for cereal matrices and dry foods [88,89,90].

### 3.5. Texture

Hardness and crispness are textural properties that attract significant attention in the evaluation of baked goods because of their close association with human perceptions of freshness. These parameters should be as low as possible [91]. The hardest (6.03 ± 0.14 N) and the least crispness (0.479 ± 0.027 mm) is the control sample. The increasing addition of carp meat results in a decrease in the hardness of the tested breadsticks (Table 8). The smallest addition of carp meat (5%) caused an approximately 40% decrease in the hardness of the breadsticks. The smallest addition of carp meat (5%) caused an approximately 41% decrease in the hardness of the breadsticks (3.53 ± 0.05 N), while the largest addition (30%) contributed to an over 53% decrease in hardness (2.82 *±* 0.02 N) compared to the control sample. The hardness did not differ significantly (*p* > 0.05) in samples S10–S25, regardless of the carp meat used in the formulas; however, with increasing amounts of carp meat, the hardness decreases slightly from 3.06 ± 0.19 N in the sample S10 up to 2.94 ± 0.08 N in the S25 sample. The increasing addition of carp meat reduces the hardness of the snacks while increasing their crispness. The increasing addition of carp meat reduces the hardness of the snacks while increasing their crispness, with all samples differing statistically significantly in crispness. This relationship can be explained by the reaction between carbohydrates and proteins. During baking, cross-links form between the swollen starch and denatured gluten. The more swollen the starch granules, the more their number increases, thus increasing the surface area of potential contact with protein. These cross-links are hydrogen bonds, which weaken with increasing temperature, meaning they are significantly destroyed during baking. Replacing flour with fish meat weakens the formation of the gluten network in the dough and, as a result, the breadsticks are more fracturable [92].

### 3.6. Sensory Evaluation

Sensory attributes of the breadsticks with different carp meat content are presented in Table 9. The appearance evaluation results are similar for all samples (5.0), except for S25 and S30, which receive slightly lower notes (4.5). The addition of fish has no statistically significant effect on color, and notes consistently remain around 5.0, indicating that enrichment does not negatively impact visual appeal. Aroma notes are highest in samples S5 and S10. Flavor notes follow a similar pattern, peaking at S15 (5.0) and then declining at higher enrichment levels. Texture notes remain high up to 15% carp meat addition, then decline. This reflects the effect of fish meat on the dough structure, which reduces crispness.

Overall acceptability is highest in the S10 and S15 samples (8.7 and 8.8, respectively), significantly higher than in the control sample (8.0) and S30 (6.8). These results suggest that a moderate carp content (10–25%) is most beneficial from a consumer’s point of view. Figure 4 shows photographs of S15 breadsticks, which are characterized by the best overall acceptability.

Detailed flavor profiling (Figure 5) reveals that enrichment introduces mild fishy notes that gradually increase from S5 (0.5) to S30 (2.0). Saltiness is consistent across all samples, while bitterness and sweetness remain minimal. Fatty mouthfeel also increases slightly with the addition of carp, consistent with the higher fat content in the enriched samples.

Aroma profiling reflects flavor development. The fishy aroma is absent in the control sample but intensified with enrichment, reaching 2.0 in S30. Although the mild fishy aroma is generally acceptable, excessive intensity at higher levels likely contributes to lower sensory notes. Importantly, undesirable notes such as bland, bitter, and sour are minimal, suggesting that the addition of carp did not cause significant negative sensory deficiencies.

These results are consistent with previous studies that show that moderate addition of fish or seafood ingredients to cereal products or baked goods can improve sensory appeal by providing umami and savory notes, but excessive addition can lead to a strong fishy flavor and reduced consumer acceptance [93,94,95].

## 4. Conclusions

Enriching breadsticks with carp meat has successfully transformed a traditionally low-value cereal product into a functional food with enhanced nutritional and bioactive potential. These studies demonstrate that fish enrichment not only improves protein quality and the presence of health-promoting compounds but also contributes to oxidative stability and acceptable sensory properties when used in optimal quantities. Carp enrichment resulted in substantial improvements in protein quantity and quality, with total amino acids increasing by up to 44% and lysine—the primary limiting amino acid in cereal products—rising almost seven-fold in the highest substitution level. Importantly, the retention of available lysine after baking improved markedly, with losses reduced from 40% in the control to approximately 17% in samples enriched with 30% carp. This indicates that carp addition not only increases the total lysine pool but also protects heat-sensitive amino acids during thermal processing, thereby preserving biological value. Despite higher PUFA content, oxidative stability remained acceptable, and moderate carp addition maintained favorable sensory characteristics, confirming that enrichment fish be achieved without compromising consumer acceptance. Research demonstrates that carp-fortified breadsticks can serve as an effective vehicle for delivering high-quality proteins, indispensable amino acids, omega-3 fatty acids, and bioavailable minerals in a format that is familiar, convenient, and widely consumed. This approach offers a practical strategy to increase fish while supporting the development of innovative, functional snack products.

## Figures and Tables

**Figure 1 foods-14-04066-f001:**
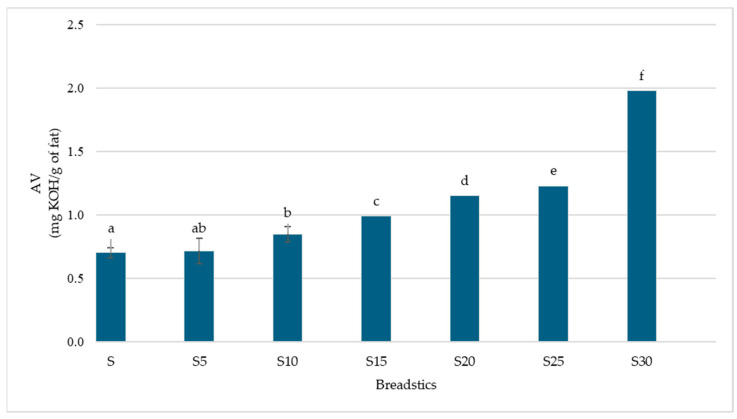
The acid value of breadsticks with different carp meat additions. Means in a bar with the same lowercase letter do not differ significantly (*p* < 0.05).

**Figure 2 foods-14-04066-f002:**
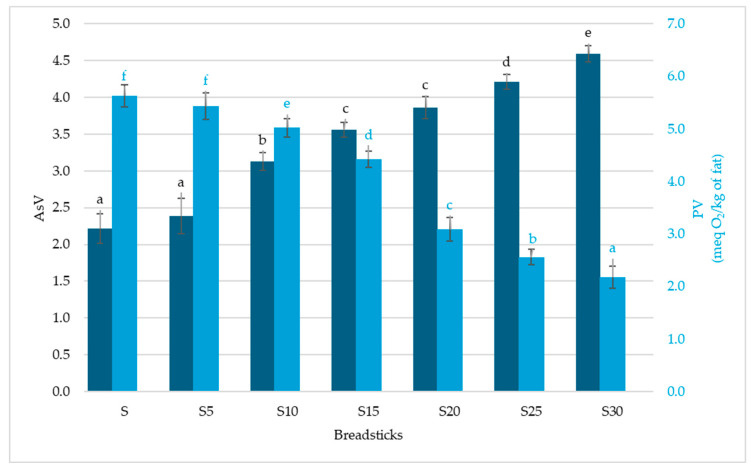
The anisidine value and peroxide value of breadsticks with different carp meat additions. Means in a bar with the same lowercase letter do not differ significantly (*p* < 0.05).

**Figure 3 foods-14-04066-f003:**
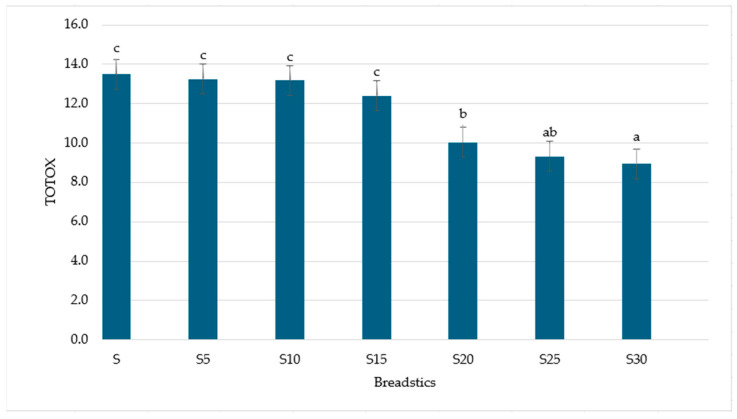
TOTOX of breadsticks with different carp meat additions. Means in a bar with the same lowercase letter do not differ significantly (*p* < 0.05).

**Figure 4 foods-14-04066-f004:**
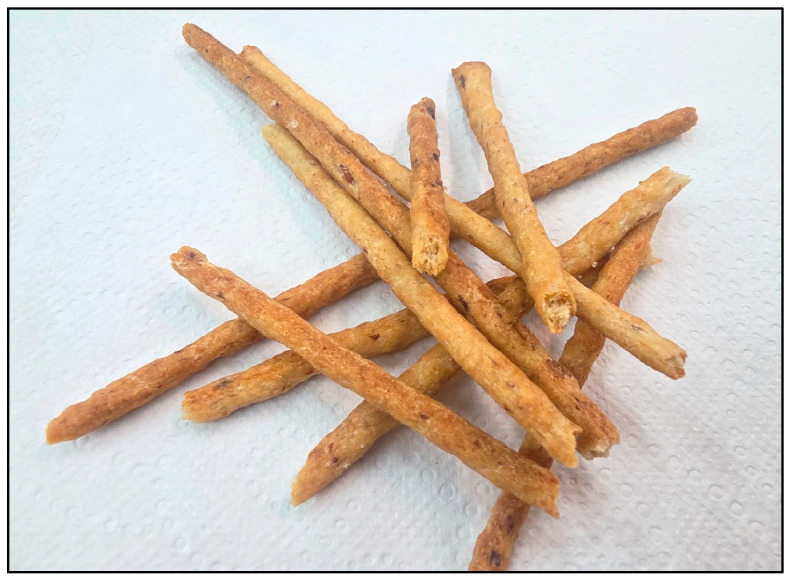
Breadsticks with 15% carp meat addition—S15 (photography by Grzegorz Tokarczyk).

**Figure 5 foods-14-04066-f005:**
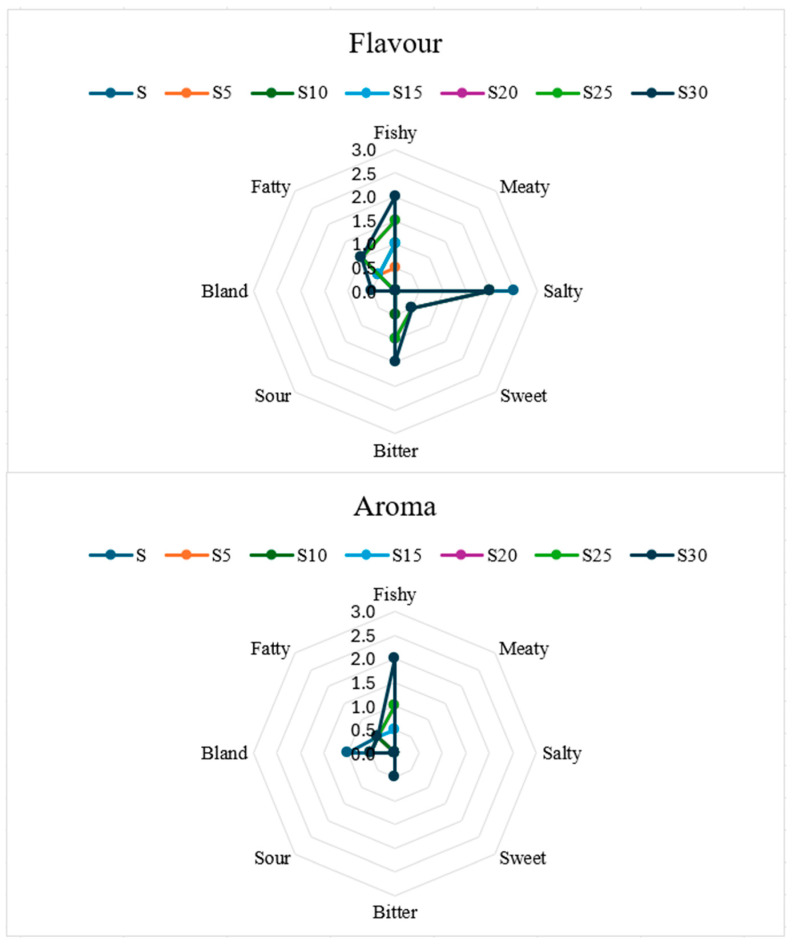
Flavor and aroma profiling of breadsticks with different carp meat additions.

**Table 1 foods-14-04066-t001:** Ingredients used to prepare breadsticks snacks with various additions of carp meat [g].

	Breadsticks						
	S	S5	S10	S15	S20	S25	S30
Flour	132.5	130.0	127.5	125.0	122.5	120.0	117.5
Oil	35.0	35.0	35.0	35.0	35.0	35.0	35.0
Water	70.0	60.0	50.0	40.0	30.0	20.0	10.0
Yeast	5.0	5.0	5.0	5.0	5.0	5.0	5.0
Baking powder	2.5	2.5	2.5	2.5	2.5	2.5	2.5
Salt	2.5	2.5	2.5	2.5	2.5	2.5	2.5
Sugar	2.5	2.5	2.5	2.5	2.5	2.5	2.5
Carp meat	0.0	12.5	25.0	37.5	50.0	62.5	70.0

S—breadsticks without carp meat; S5—breadsticks with 5% carp meat; S10—breadsticks with 10% carp meat; S15—breadsticks with 15% carp meat; S20—breadsticks with 20% carp meat; S25—breadsticks with 25% carp meat; S30—breadsticks with 30% carp meat.

**Table 2 foods-14-04066-t002:** The basic composition of carp meat and breadsticks with different carp meat additions [g/100 g dry weight].

	Carp	Breadsticks						
	Meat	S	S5	S10	S15	S20	S25	S30
Dry matter	24.98 ± 0.22	94.24 ± 0.25 ^d^	94.14 ± 0.18 ^d^	94.04 ± 0.26 ^cd^	93.95 ± 0.11 ^c^	93.76 ± 0.12 ^b^	93.64 ± 0.26 ^ab^	93.51 ± 0.12 ^a^
Fat	4.32 ± 0.12	18.02 ± 0.08 ^a^	18.46 ± 0.27 ^b^	19.25 ± 0.16 ^c^	20.40 ± 10 ^d^	21.53 ± 0.15 ^e^	22.27 ± 0.12 ^f^	23.16 ± 0.16 ^g^
Protein	18.55 ± 0.09	9.73 ± 0.16 ^a^	11.28 ± 0.20 ^b^	14.19 ± 0.15 ^c^	15.59 ± 0.10 ^d^	17.60 ± 0.12 ^e^	18.62 ± 0.09 ^f^	19.52 ± 0.11 ^g^
Ash	0.66 ± 0.06	4.18 ± 0.04 ^g^	4.05 ± 0.03 ^fs^	3.40 ± 0.11 ^e^	3.03 ± 0.05 ^d^	2.55 ± 0.06 ^c^	2.32 ± 0.05 ^b^	2.21 ± 0.02 ^a^
Fiber	-	1.97 ± 0.03 ^d^	1.55 ± 0.01 ^c^	1.00 ± 0.05 ^ab^	0.98 ± 0.02 ^a^	1.06 ± 0.04 ^b^	1.03 ± 0.03 ^b^	1.04 ± 0.10 ^ab^
NFE	-	60.34 ± 0.12 ^g^	58.80 ± 0.15 ^f^	56.20 ± 0.10 ^e^	53.95 ± 0.11 ^d^	51.02 ± 0.10 ^c^	49.40 ± 0.12 ^b^	47.58 ± 0.15 ^a^

Mean values of breadsticks in rows with the same lowercase letter do not differ significantly (*p* < 0.05). NFE—nitrogen-free extract.

**Table 3 foods-14-04066-t003:** Content of microminerals of carp meat and breadsticks with different carp meat additions.

	Carp	Breadsticks						
	Meat	S	S5	S10	S15	S20	S25	S30
P [g/kg]	2.45 ± 0.34	3.89 ± 0.11 ^f^	2.20 ± 0.12 ^a^	2.15 ± 0.10 ^a^	2.36 ± 0.05 ^b^	2.55 ± 0.00 ^c^	2.76 ± 0.09 ^d^	2.92 ± 0.10 ^e^
K [g/kg]	3.95 ± 0.19	1.60 ± 0.05 ^a^	1.85 ± 0.05 ^b^	2.02 ± 0.05 ^c^	2.16 ± 0.06 ^d^	2.30 ± 0.04 ^e^	2.59 ± 0.09 ^f^	2.72 ± 0.02 ^g^
Mg [g/kg]	1.15 ± 0.09	185.20 ± 1.18 ^a^	196.16 ± 1.19 ^b^	208.05 ± 2.13 ^c^	216.20 ± 2.11 ^d^	228.21 ± 3.16 ^e^	231.12 ± 2.12 ^f^	232.11 ± 1.50 ^f^
Na [g/kg]	0.52 ± 0.06	5.38 ± 0.15 ^a^	5.45 ± 0.10 ^a^	5.25 ± 0.10 ^a^	5.40 ± 0.10 ^a^	5.22 ± 0.20 ^a^	5.36 ± 0.10 ^a^	5.25 ± 0.10 ^a^
Fe [mg/kg]	7.32 ± 0.22	20.24 ± 0.10 ^a^	21.95 ± 0.22 ^d^	22.32 ± 0.16 ^e^	22.02 ± 0.11 ^d^	21.68 ± 0.05 ^c^	21.46 ± 0.11 ^b^	21.55 ± 0.04 ^b^
Ca [mg/kg]	311.20 ± 0.68	18.21 ± 0.05 ^g^	11.63 ± 0.21 ^a^	12.06 ± 0.10 ^b^	12.86 ± 0.04 ^c^	14.86 ± 0.12 ^d^	15.32 ± 0.06 ^e^	17.85 ± 0.05 ^f^
Ni [mg/kg]	-	-	-	-	-	-	-	-
Cr [mg/kg]	0.04 ± 0.01	0.04 ± 0.01 ^a^	0.04 ± 0.01 ^a^	0.04 ± 0.02 ^a^	0.04 ± 0.02 ^a^	0.03 ± 0.00 ^a^	0.03 ± 0.00 ^a^	0.03 ± 0.01 ^a^
Pb [mg/kg]	-	-	-	-	-	-	-	-
Cd [mg/kg]	-	-	-	-	-	-	-	-
Mn [mg/kg]	0.76 ± 0.12	4.02 ± 0.11 ^a^	5.12 ± 0.23 ^d^	5.05 ± 0.23 ^cd^	4.95 ± 0.15 ^c^	4.45 ± 0.26 ^b^	4.52 ± 0.16 ^b^	4.66 ± 0.10 ^b^
Co [mg/kg]	-	-	-	-	-	-	-	-
Cu [mg/kg]	-	-	-	-	-	-	-	-
Zn [mg/kg]	3.42 ± 0.33	8.20 ± 0.23 ^a^	8.52 ± 0.21 ^b^	8.48 ± 0.10 ^b^	8.45 ± 0.15 ^b^	8.35 ± 0.05 ^a^	8.89 ± 0.10 ^c^	9.24 ± 0.05 ^d^

Mean values of breadsticks in rows with the same lowercase letter do not differ significantly (*p* < 0.05).

**Table 4 foods-14-04066-t004:** Antioxidant activity of carp meat and breadsticks with different carp meat additions.

	Carp	Breadsticks						
	Meat	S	S5	S10	S15	S20	S25	S30
TEAC [µmol TE/g]	2.25 ± 0.10	1.71 ± 0.18 ^a^	1.84 ± 0.05 ^a^	1.94 ± 0.04 ^a^	2.23 ± 0.06 ^b^	2.62 ± 0.14 ^c^	3.12 ± 0.04 ^d^	3.50 ± 0.26 ^e^
FRAP [µmol TE/g]	28.36 ± 0.60	68.7 ± 1.2 ^a^	75.2 ± 0.9 ^b^	80.1 ± 0.7 ^c^	86.3 ± 1.2 ^d^	89.9 ± 3.6 ^e^	93.2 ± 0.5 ^ed^	94.7 ± 2.2 ^d^
DPPH [µmol TE/g]	0.312 ± 0.015	0.516 ± 0.031 ^a^	0.603 ± 0.012 ^b^	0.636 ± 0.018 ^c^	0.675 ± 0.015 ^c^	0.680 ± 0.017 ^c^	0.722 ± 0.016 ^d^	0.738 ± 0.021 ^d^
TPC [mg GAE/g]	0.569 ± 0.021	0.678 ± 0.009 ^d^	0.612 ± 0.010 ^a^	0.615 ± 0.008 ^a^	0.625 ± 0.012 ^b^	0.637 ± 0.010 ^b^	0.659 ± 0.012 ^c^	0.673 ± 0.028 ^d^

Means values of breadsticks in rows with the same lowercase letter do not differ significantly (*p* < 0.05). TEAC–Trolox equivalent antioxidant capacity; FRAP–Ferric reducing antioxidant power; DPPH–Radical scavenging activity; TPC–Total phenolic compounds.

**Table 5 foods-14-04066-t005:** Fatty acid profile of carp meat breadsticks with different carp meat additions [%].

	Carp	Breadsticks						
	Meat	S	S5	S10	S15	S20	S25	S30
C10:0	-	0.01 ± 0.00 ^a^	0.01 ± 0.01 ^a^	0.01 ± 0.00 ^a^	0.01 ± 0.00 ^a^	0.01 ± 0.00 ^a^	0.01 ± 0.01 ^a^	0.01 ± 0.00 ^a^
C12:0	0.07 ± 0.02	0.01 ± 0.00 ^a^	0.01 ± 0.00 ^a^	0.01 ± 0.01 ^a^	0.01 ± 0.00 ^a^	0.01 ± 0.00 ^a^	0.01 ± 0.00 ^a^	0.01 ± 0.01 ^a^
C14:0	0.80 ± 0.01	0.07 ± 0.03 ^a^	0.11 ± 0.02 ^b^	0.14 ± 0.02 ^c^	0.14 ± 0.02 ^c^	0.10 ± 0.01 ^ab^	0.15 ± 0.01 ^c^	0.15 ± 0.0 ^c^
C15:0	0.67 ± 0.00	0.03 ± 0.02 ^a^	0.03 ± 0.01 ^a^	0.04 ± 0.00 ^a^	0.03 ± 0.01 ^a^	0.02 ± 0.02 ^a^	0.03 ± 0.01 ^a^	0.04 ± 0.0 ^a^
C16:0	14.94 ± 0.53	6.29 ± 0.02 ^a^	6.67 ± 0.04 ^b^	6.82 ± 0.04 ^c^	6.85 ± 0.03 ^cd^	6.87 ± 0.04 ^de^	6.91 ± 0.03 ^e^	6.89 ± 0.02 ^e^
C16:1	2.42 ± 0.24	0.27 ± 0.05 ^a^	0.51 ± 0.03 ^b^	0.70 ± 0.05 ^c^	0.74 ± 0.03 ^c^	0.81 ± 0.03 ^d^	0.85 ± 0.02 ^d^	0.92 ± 0.01 ^e^
C17:0	0.25 ± 0.13	0.07 ± 0.02 ^a^	0.08 ± 0.01 ^a^	0.09 ± 0.03 ^a^	0.08 ± 0.02 ^a^	0.08 ± 0.00 ^a^	0.08 ± 0.01 ^a^	0.09 ± 0.04 ^a^
C17:1	0.25 ± 0.11	0.08 ± 0.02 ^a^	0.10 ± 0.01 ^a^	0.10 ± 0.03 ^a^	0.09 ± 0.00 ^a^	0.10 ± 0.02 ^a^	0.08 ± 0.01 ^a^	0.10 ± 0.00 ^a^
C18:0	5.18 ± 0.10	2.91 ± 0.06 ^c^	3.31 ± 0.03 ^f^	3.22 ± 0.05 ^e^	3.06 ± 0.04 ^d^	3.08 ± 0.02 ^d^	2.42 ± 0.03 ^b^	2.19 ± 0.04 ^a^
C18:1 ω9	44.57 ± 0.89	56.78 ± 0.10 ^d^	55.61 ± 0.12 ^a^	55.86 ± 0.15 ^b^	56.32 ± 0.11 ^c^	56.50 ± 0.20 ^c^	57.10 ± 0.12 ^e^	57.32 ± 0.21 ^e^
C18:2 ω6	17.48 ± 0.23	20.52 ± 0.15 ^d^	20.27 ± 0.12 ^c^	19.90 ± 0.10 ^b^	19.94 ± 0.10 ^b^	19.78 ± 0.10 ^a^	19.85 ± 0.15 ^ab^	19.86 ± 0.16 ^ab^
C20:0	0.27 ± 0.05	1.04 ± 0.05 ^c^	1.01 ± 0.03 ^c^	1.01 ± 0.04 ^c^	0.93 ± 0.05 ^b^	0.89 ± 0.06 ^ab^	0.85 ± 0.05 ^a^	0.78 ± 0.10 ^a^
C18:3 ω6	0.40 ± 0.06	0.28 ± 0.04 ^a^	0.33 ± 0.02 ^b^	0.28 ± 0.03 ^a^	0.28 ± 0.02 ^a^	0.28 ± 0.03 ^a^	0.27 ± 0.03 ^a^	0.27 ± 0.02 ^a^
C18:3 ω3	2.48 ± 0.15	8.61 ± 0.12 ^b^	8.42 ± 0.11 ^a^	8.39 ± 0.09 ^a^	8.40 ± 0.12 ^a^	8.41 ± 0.10 ^a^	8.38 ± 0.10 ^a^	8.37 ± 0.11 ^a^
C20:1 ω9	2.80 ± 0.12	1.59 ± 0.12 ^b^	1.44 ± 0.11 ^a^	1.71 ± 0.05 ^c^	1.61 ± 0.08 ^b^	1.65 ± 0.10 ^bc^	1.68 ± 0.06 ^bc^	1.65 ± 0.05 ^b^
C18:4 ω3	0.14 ± 0.01	0.03 ± 0.02 ^a^	0.04 ± 0.00 ^a^	0.03 ± 0.01 ^a^	0.03 ± 0.02 ^a^	0.03 ± 0.01 ^a^	0.04 ± 0.01 ^a^	0.05 ± 0.02 ^a^
C20:2 ω6	0.49 ± 0.06	0.08 ± 0.01 ^a^	0.11 ± 0.00 ^b^	0.11 ± 0.01 ^b^	0.10 ± 0.01 ^b^	0.12 ± 0.02 ^b^	0.13 ± 0.02 ^b^	0.13 ± 0.03 ^b^
C22:0	0.25 ± 0.02	0.55 ± 0.02 ^c^	0.52 ± 0.01 ^b^	0.53 ± 0.03 ^bc^	0.50 ± 0.00 ^b^	0.50 ± 0.02 ^b^	0.45 ± 0.02 ^a^	0.46 ± 0.03 ^a^
C22:1 ω9	0.25 ± 0.01	0.00 ± 0.00 ^a^	0.51 ± 0.03 ^f^	0.35 ± 0.01 ^e^	0.25 ± 0.03 ^d^	0.20 ± 0.02 ^c^	0.18 ± 0.02 ^bc^	0.15 ± 0.04 ^b^
C20:4 ω6	2.74 ± 0.06	0.31 ± 0.04 ^c^	0.03 ± 0.01 ^a^	0.05 ± 0.02 ^a^	0.05 ± 0.01 ^ab^	0.05 ± 0.00 ^b^	0.04 ± 0.01 ^a^	0.05 ± 0.02 ^ab^
C23:0	-	0.04 ± 0.01 ^c^	0.04 ± 0.01 ^c^	0.03 ± 0.00 ^b^	0.03 ± 0.01 ^bc^	0.02 ± 0.01 ^ab^	0.02 ± 0.01 ^ab^	0.01 ± 0.00 ^a^
C20:5 ω3	0.35 ± 0.03	-	0.03 ± 0.00 ^a^	0.05 ± 0.00 ^b^	0.06 ± 0.02 ^b^	0.05 ± 0.00 ^b^	0.05 ± 0.01 ^b^	0.06 ± 0.01 ^b^
C24:0	-	0.23 ± 0.04 ^de^	0.37 ± 0.02 ^f^	0.25 ± 0.02 ^e^	0.19 ± 0.03 ^d^	0.15 ± 0.01 ^c^	0.11 ± 0.02 ^b^	0.06 ± 0.01 ^a^
C24:1 ω9	0.11 ± 0.01	0.19 ± 0.01 ^d^	0.33 ± 0.05 ^e^	0.21 ± 0.03 ^d^	0.16 ± 0.02 ^c^	0.14 ± 0.02 ^c^	0.10 ± 0.01 ^b^	0.05 ± 0.00 ^a^
C22:4 ω6	0.21 ± 0.02	-	0.04 ± 0.02 ^a^	0.03 ± 0.00 ^a^	0.03 ± 0.01 ^a^	0.03 ± 0.00 ^a^	0.03 ± 0.01 ^a^	0.02 ± 0.00 ^a^
C22:5 ω3	0.31 ± 0.06	-	0.03 ± 0.01 ^ab^	0.02 ± 0.00 ^a^	0.04 ± 0.00 ^bc^	0.04 ± 0.01 ^bc^	0.05 ± 0.01 ^c^	0.11 ± 0.01 ^d^
C22:6 ω3	2.56 ± 0.10	-	0.02 ± 0.00 ^a^	0.05 ± 0.01 ^b^	0.06 ± 0.01 ^b^	0.08 ± 0.00 ^c^	0.13 ± 0.02 ^d^	0.20 ± 0.03 ^e^
SFA	22.45 ± 0.16	11.25 ± 0.12 ^c^	12.17 ± 0.13 ^e^	12.15 ± 0.11 ^e^	11.84 ± 0.09 ^d^	11.73 ± 0.18 ^d^	11.04 ± 0.10 ^b^	10.69 ± 0.16 ^a^
MUFA	50.39 ± 0.23	58.92 ± 1.02 ^a^	58.50 ± 1.12 ^ab^	58.94 ± 0.93 ^ab^	59.16 ± 0.86 ^ab^	59.40 ± 0.63 ^ab^	59.99 ± 0.78 ^bc^	60.19 ± 0.67 ^c^
PUFA	27.16 ± 0.21	29.83 ± 0.26 ^c^	29.33 ± 0.22 ^b^	28.91 ± 0.19 ^ab^	28.99 ± 0.15 ^ab^	28.87 ± 0.23 ^a^	28.97 ± 0.30 ^ab^	29.12 ± 0.21 ^b^
ω3/ω6	0.27	0.41	0.41	0.42	0.42	0.42	0.43	0.43
DHA/EPA	7.31	-	0.68	0.94	1.00	1.60	2.60	3.33

Means values of breadsticks in rows with the same lowercase letter do not differ significantly (*p* < 0.05). SFA–Saturated fatty acids; MUFA–Monounsaturated fatty acids; PUFA–Polyunsaturated fatty acids; DHA–Docosahexaenoic acid; EPA–Eicosapentaenoic acid.

**Table 6 foods-14-04066-t006:** Amino acid profile of carp meat and breadsticks with different carp meat additions [g/100 g wet weight].

	Carp Meat	Breadsticks						
		S	S5	S10	S15	S20	S25	S30
Alanine *	1110.0 ± 12.0	440.0 ± 25.0 ^c^	430.0 ± 15.0 ^bc^	400.0 ± 10.0 ^a^	425.0 ± 11.0 ^bc^	400.0 ± 15.0 ^ab^	415.0 ± 11.0 ^b^	430.0 ± 20.0 ^bc^
Arginine *	1026.0 ± 32.0	730.0 ± 23.0 ^a^	810.0 ± 13.0 ^b^	850.0 ± 20.0 ^c^	890.0 ± 10.0 ^f^	962.0 ± 21.0 ^e^	1084.0 ± 21.0 ^f^	1106.0 ± 32.0 ^f^
Cysteine *	275.0 ± 19.0	251.0 ± 12.0 ^c^	235.0 ± 10.0 ^a^	240.0 ± 10.0 ^ab^	250.0 ± 10.0 ^bc^	265.0 ± 10.0 ^d^	280.0 ± 15.0 ^de^	290.0 ± 11.0 ^e^
Cystine	188.0 ± 15.0	240.0 ± 29.0 ^ab^	240.0 ± 10.0 ^a^	244.0 ± 12.0 ^a^	251.0 ± 11.0 ^b^	263.0 ± 15.0 ^bc^	277.0 ± 12.0 ^cd^	290.2 ± 21.0 ^d^
Phenylalanine	855.0 ± 22.0	350.0 ± 31.0 ^a^	442.0 ± 12.00 ^b^	580.0 ± 32.0 ^c^	602.0 ± 16.0 ^c^	699.0 ± 12.0 ^d^	786.0 ± 19.0 ^e^	822.0 ± 26.0 ^f^
Glycine *	732.0 ± 10.0	240.0 ± 21.0 ^a^	265.0 ± 13.0 ^b^	288.0 ± 20.0 ^c^	312.0 ± 16.0 ^d^	366.0 ± 14.0 ^e^	402.0 ± 10.0 ^f^	430.0 ± 26.0 ^g^
Histidine	841.0 ± 23.0	652.0 ± 34.0 ^b^	632.0 ± 20.0 ^ab^	613.0 ± 21.0 ^a^	698.0 ± 12.0 ^c^	745.0 ± 18.0 ^d^	803.0 ± 0.11 ^e^	820.0 ± 12.0 ^f^
Hydroxyproline *	243.0 ± 11.0	120.0 ± 14.0 ^a^	110.0 ± 11.0 ^a^	120.0 ± 15.0 ^a^	126.0 ± 10.0 ^a^	156.0 ± 11.0 ^b^	195.0 ± 13.0 ^c^	201.0 ± 16.0 ^c^
Isoleucine	1203.0 ± 30.0	640.0 ± 26.0 ^a^	653.0 ± 19.0 ^a^	660.0 ± 23.0 ^a^	698.0 ± 13.0 ^b^	746.0 ± 13.0 ^c^	829.0 ± 19.0 ^d^	881.0 ± 29.0 ^e^
Aspartic acid *	1451.0 ± 35.0	820.0 ± 36.0 ^a^	974.0 ± 21.0 ^b^	1033.0 ± 23.0 ^c^	1096.0 ± 23.0 ^d^	1129.0 ± 0.16 ^e^	1185.0 ± 0.21^f^	1200.0 ± 32.0 ^f^
Glutamic acid *	2982.0 ± 23.0	1581.0 ± 21.0 ^a^	1650.0 ± 23.0 ^b^	1870.0 ± 14.0 ^e^	1762.0 ± 26.0 ^d^	1687.0 ± 25.0 ^c^	1555.0 ± 29.0 ^a^	1582.0 ± 38.0 ^a^
Leucine	1835.0 ± 18.0	623.0 ± 29.0 ^a^	721.0 ± 19.0 ^b^	940.0 ± 13.0 ^c^	946.0 ± 10.0 ^c^	957.0 ± 13.0 ^c^	953.0 ± 18.0 ^c^	980.0 ± 24.0 ^d^
Lysine	1778.0 ± 13.0	143.0 ± 18.0 ^a^	213.0 ± 11.0 ^b^	325.0 ± 19.0 ^c^	523.0 ± 21.0 ^d^	798.0 ± 19.0 ^e^	886.0 ± 26.0 ^f^	980.0 ± 36.0 ^g^
Methionine	726.0 ± 28.0	110.0 ± 11.0 ^a^	123.0 ± 15.0 ^ab^	131.0 ± 10.0 ^b^	136.0 ± 10.0 ^b^	156.0 ± 11.0 ^c^	168.0 ± 16.0 ^c^	172.0 ± 21.0 ^c^
Proline *	830.0 ± 13.0	1016.0 ± 28.0 ^a^	1010.0 ± 15.0 ^a^	1005.0 ± 10.0 ^a^	995.0 ± 21.0 ^ab^	986.0 ± 09.0 ^b^	968.0 ± 13.0 ^c^	970.0 ± 15.0 ^c^
Serine *	869.0 ± 21.0	580.0 ± 21.0 ^a^	650.0 ± 25.0 ^b^	700.0 ± 28.0 ^c^	786.0 ± 18.0 ^d^	821.0 ± 20.0 ^e^	902.0 ± 10.0 ^f^	920.0 ± 10.0 ^g^
Taurine (total) *	-	-	-	-	-	-	-	-
Threonine	836.0 ± 23.0	330.0 ± 21.0 ^a^	356.0 ± 19.0 ^b^	370.0 ± 21.0 ^b^	423.0 ± 13.0 ^c^	516.0 ± 20.0 ^d^	602.0 ± 0.20 ^e^	616.0 ± 11.0 ^e^
Tryptophan	292.0 ± 16.0	302.0 ± 32.0 ^a^	301.0 ± 21.0 ^a^	300.0 ± 29.0 ^a^	303.0 ± 21.0 ^a^	305.0 ± 16.0 ^a^	307.0 ± 10.0 ^a^	310.0 ± 10.0 ^a^
Tyrosine *	896.0 ± 10.0	500.0 ± 20.0 ^a^	620.0 ± 25.0 ^b^	651.0 ± 21.0 ^b^	720.0 ± 10.0 ^c^	790.0 ± 25.0 ^d^	815.0 ± 21.0 ^d^	914.0 ± 22.0 ^e^
Valine	1322.0 ± 19.0	651.0 ± 23.0 ^a^	698.0 ± 23.0 ^b^	720.0 ± 26.0 ^b^	752.0 ± 28.0 ^c^	777.0 ± 20.0 ^c^	826.0 ± 18.0 ^d^	890.0 ± 23.0 ^d^
TAA	20,290.0 ± 26.0	10,319.0 ± 19.0 ^a^	11,343.0 ± 15.0 ^b^	12,040.0 ± 21.0 ^c^	12,694.0 ± 12.0 ^d^	14,126.0 ± 22.0 ^e^	14,238.0 ± 15.0 ^f^	14,804.0 ± 32.0 ^g^
EAA	9876.0 ± 23.0	4041.0 ± 12.0 ^a^	4379.0 ± 16.0 ^b^	4883.0 ± 13.0 ^c^	5332.0 ± 10.0 ^d^	6562.0 ± 16.0 ^f^	6437.0 ± 14.0 ^e^	6761.0 ± 20.0 ^g^
NEAA *	10,414.0 ± 30.0	6278.0 ± 22.0 ^a^	6964.0 ± 10.0 ^b^	7157.0 ± 16.0 ^c^	7362.0 ± 31.0 ^d^	7562.0 ± 12.0 ^e^	7801.0 ± 21.0 ^f^	8043.0 ± 21.0 ^g^

Means values of breadsticks in rows with the same lowercase letter do not differ significantly (*p* < 0.05). TAA—total amino acids; EAA—essential amino acids; NEAA—non-essential amino acids; *—amino acids marked with an asterisk are non-essential amino acids (NEAA*).

**Table 7 foods-14-04066-t007:** Content and losses of available lysine in breadsticks with different carp meat additions.

	Breadsticks						
	S	S5	S10	S15	S20	S25	S30
Before baking [mg/100 g]	113.0 ± 10.0 ^a^	126.0 ± 12.0 ^b^	230.0 ± 20.0 ^c^	415.0 ± 21.0 ^d^	688.0 ± 22.0 ^e^	781.0 ± 28.0 ^f^	887.0 ± 25.0 ^g^
After baking [mg/100 g]	68.0 ± 3.0 ^a^	95.0 ± 5.0 ^b^	182.0 ± 14.0 ^c^	334.0 ± 16.0 ^d^	557.0 ± 18.0 ^e^	648.0 ± 16.0 ^f^	734.0 ± 20.0 ^g^
Losses of available lysine [%]	40.0 ± 6.0 ^d^	24.4 ± 3.0 ^c^	21.0 ± 2.8 ^b^	19.6 ± 2.0 ^b^	19.0 ± 3.0 ^ab^	17.0 ± 2.2 ^a^	17.3 ± 2.0 ^a^

Means in rows with the same lowercase letter do not differ significantly (*p* < 0.05).

**Table 8 foods-14-04066-t008:** Texture parameters in breadsticks with different carp meat additions.

	Breadsticks						
	S	S5	S10	S15	S20	S25	S30
Hardness [N]	6.03 ± 0.14 ^a^	3.53 ± 0.05 ^b^	3.06 ± 0.19 ^c^	3.03 ± 0.10 ^c^	2.98 ± 0.06 ^cd^	2.94 ± 0.08 ^d^	2.82 ± 0.02 ^e^
Crispness [mm]	0.479 ± 0.027 ^a^	0.356 ± 0.015 ^b^	0.339 ± 0.026 ^c^	0.316 ± 0.016 ^d^	0.295 ± 0.010 ^e^	0.276 ± 0.009 ^f^	0.252 ± 0.012 ^g^

Means in rows with the same lowercase letter do not differ significantly (*p* < 0.05).

**Table 9 foods-14-04066-t009:** Sensory attributes of breadsticks with different carp meat additions.

	Breadsticks						
	S	S5	S10	S15	S20	S25	S30
Appearance	5.0 ± 0.0 ^b^	5.0 ± 0.0 ^b^	5.0 ± 0.0 ^b^	5.0 ± 0.0 ^b^	5.0 ± 0.0 ^b^	4.5.0 ± 0.0 ^a^	4.5.0 ± 0.5 ^ab^
Color	5.0 ± 0.0 ^b^	4.8 ± 0.3 ^b^	4.8 ± 0.3 ^b^	5.0 ± 0.0 ^b^	5.0 ± 0.5 ^ab^	4.5 ± 0.0 ^a^	4.5 ± 0.0 ^a^
Aroma	4.3 ± 0.0 ^b^	4.8 ± 0.2 ^c^	5.0 ± 0.0 ^d^	4.5 ± 0.5 ^cd^	4.0 ± 0.0 ^a^	4.3 ± 0.2 ^bc^	4.0 ± 0.0 ^a^
Flavor	4.3 ± 0.2 ^bc^	4.5 ± 0.0 ^c^	5.0 ± 0.0 ^d^	5.0 ± 0.0 ^d^	4.5 ± 0.0 ^c^	4.0 ± 0.2 ^a^	4.3 ± 0.0 ^b^
Texture	5.0 ± 0.0 ^c^	5.0 ± 0.0 ^c^	5.0 ± 0.0 ^c^	4.8 ± 0.3 ^bc^	4.5 ± 0.0 ^b^	4.2 ± 0.2 ^a^	4.0 ± 0.0 ^a^
Overall acceptability	8.0 ± 0.5 ^cd^	8.5 ± 0.5 ^d^	8.7 ± 0.2 ^d^	8.8 ± 0.0 ^d^	7.5 ± 0.5 ^bc^	7.0 ± 0.0 ^ab^	6.8 ± 0.2 ^a^

Means in rows with the same lowercase letter do not differ significantly (*p* < 0.05).

## Data Availability

The original contributions presented in the study are included in the article. Further inquiries can be directed to the corresponding author.

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
