# Peer review of "Nutritional Composition, Bioactive Properties, and Sensory Evaluation of Breadsticks Enriched with Carp Meat (Cyprinus carpio, L.)"

_foods, 2025, doi:10.3390/foods14234066_

Round 1
Reviewer 1 Report
Comments and Suggestions for Authors
- The introduction section is overly generalized and lacks sufficient focus on the specific material used for fortification. The authors should clearly discuss the fish species (Cyprinus carpio, L.) used in the study, including relevant background on its nutritional composition, classification as an oily or lean fish, and its potential as a fortifying ingredient for bakery products.
- The introduction lacks sufficient citation support regarding previous research on marine- or fish-based fortification of commonly consumed snacks. The authors are advised to include relevant and up-to-date literature references that discuss the incorporation of marine-derived ingredients (such as fish meat, fish protein hydrolysates, seaweed, or fish oils) into bakery or snack products.
- The manuscript does not clearly establish how the proposed product compares with commonly used fortified foods designed to address nutrient deficiencies in the target population. The authors should include a brief discussion or literature comparison highlighting existing fortified products (e.g., protein-enriched bakery items, micronutrient-fortified snacks, or fish-based food interventions) that are currently used to mitigate similar nutritional deficiencies.
- In Table 1, the word fish should be rephrased as fish remains
- Add the reference of methodology used in 3. content of micro and microminerals
- The study presents the nutritional composition of the fortified breadsticks; however, no detailed nutritional analysis of the carp remains used as the fortifying ingredient is provided. Conducting or including such an analysis would have offered valuable insight into the nutritional rationale behind using these remains and clarified their specific contribution to the final product’s composition.
- The manuscript states that sensory evaluation was performed by ten trained panelists; however, it does not describe how inter-panelist reliability or agreement was ensured. To validate the sensory results, the authors should clarify the procedures used to ensure consistency and reproducibility among evaluators. This may include details such as: Calculation of inter-rater reliability metrics (e.g., Cronbach’s alpha, Kendall’s coefficient of concordance, or analysis of variance among raters)
- Justify the use of parametric tests (ANOVA), was the normality distribution checked before applying this statistical model
- Table 2 presents the proximate composition of the breadsticks, including fat, protein, ash, and fiber; however, the nitrogen-free extract (NFE), which represents the carbohydrate fraction of the product — is missing. Since NFE is a key nutritional parameter in proximate analysis and contributes significantly to the energy value and overall macronutrient balance of the snack, it should be included for completeness.
- In line 241, the authors refer to phosphorus and calcium as macronutrients; however, this terminology is inaccurate. These elements should be correctly referred to as macrominerals
- The results section would benefit from including the proportionate changes in macro- and micronutrient composition between the control and fortified breadstick samples. Presenting these changes, authors can add percentage increases or decreases relative to the control.
- The authors report an increase in AV after fish remains addition; however, the manuscript should also address potential side effects, such as increased lipid peroxidation and reduced shelf life
- The authors claim that the addition of fish proteins produces antioxidant peptides, yet simply incorporating fish remains without prior enzymatic hydrolysis is unlikely to generate these bioactive peptides. The manuscript lacks any evidence from proteolytic treatment or controlled hydrolysis experiments that could support this mechanism. Furthermore, the reported antioxidant effect has not been validated in a biological system, and there is no data on the bioavailability or in vivo release of these peptides. Without such validation, it is unclear whether the product would exert the claimed antioxidant effects in the body. The discussion should also address potential limitations, including variability in peptide release and stability, as well as the possible impact on shelf life and lipid peroxidation
- Table 6 is not completely visible (should be readjusted)
- The amino acid analysis could be strengthened by comparing the results with the nutritional requirements of the target population and FAO standards, which would enhance the nutritional relevance and value of the study.
Author Response
Response to the Reviewers' comments on manuscript ID: foods-4009058
Authors would like to thank for the comments to our journal submission entitled: „Nutritional Composition, Bioactive Properties, and Sensory Evaluation of Breadsticks Enriched with Carp Meat (Cyprinus carpio, L.)”. We appreciate the astute observations and constructive comments which helped us to improve our submission.
Changes made to the manuscript are marked in red, and the response includes detailed locations of the changes.
Reviewer #1
Comment 1: The introduction section is overly generalized and lacks sufficient focus on the specific material used for fortification. The authors should clearly discuss the fish species (Cyprinus carpio, L.) used in the study, including relevant background on its nutritional composition, classification as an oily or lean fish, and its potential as a fortifying ingredient for bakery products.
Response 1: We thank the Reviewer for this valuable comment. The Introduction has been substantially expanded to provide a clearer scientific context for the use of common carp (Cyprinus carpio L.). The revised text is now more specific, focused, and aligned with the Reviewer’s expectations (1. Introduction).
C2: The introduction lacks sufficient citation support regarding previous research on marine- or fish-based fortification of commonly consumed snacks. The authors are advised to include relevant and up-to-date literature references that discuss the incorporation of marine-derived ingredients (such as fish meat, fish protein hydrolysates, seaweed, or fish oils) into bakery or snack products.
R2: Thank you for this comment. The Introduction has been revised to include up-to-date literature on the fortification of bakery and snack products with marine-derived ingredients (1. Introduction).
C3: The manuscript does not clearly establish how the proposed product compares with commonly used fortified foods designed to address nutrient deficiencies in the target population. The authors should include a brief discussion or literature comparison highlighting existing fortified products (e.g., protein-enriched bakery items, micronutrient-fortified snacks, or fish-based food interventions) that are currently used to mitigate similar nutritional deficiencies.
R3: Thank you for this valuable comment. The revised manuscript now includes examples of protein-enriched bakery products. The new content has been incorporated into the revised manuscript (1. Introduction).
C4: In Table 1, the word fish should be rephrased as fish remains
R4: Thank you for this comment. We agree that the ingredient listed in Table 1 should clearly reflect the raw material used. The carp material incorporated into breadsticks originated from edible remains of industrial filleting (backbones, collarbones, fin fragments), which were subsequently separated, cleaned, and mechanically processed to obtain clean comminuted fish meat. To accurately represent both the nature of the starting material and the form used in the formulation, we have updated Table 1 to use the term “carp meat" (Table 1).
C5: Add the reference of methodology used in 3. content of micro and microminerals
R5: Thank you for this important remark. A reference to the analytical methodology used for the determination of micro- and macromineral content has now been added in 2.3. Content of micro and macro elements (L152).
C6: The study presents the nutritional composition of the fortified breadsticks; however, no detailed nutritional analysis of the carp remains used as the fortifying ingredient is provided. Conducting or including such an analysis would have offered valuable insight into the nutritional rationale behind using these remains and clarified their specific contribution to the final product’s composition.
R6: Thank you for this insightful comment. We agree that providing nutritional information on the carp remains would strengthen the rationale for their use as a fortifying ingredient. Initially, we were guided by the fact that in our previous publications, we described the characteristics of carp, and we wanted to avoid repetition. Currently, we have expanded the manuscript to include published compositional data for common carp meat. This information clarifies the nutritional contribution of the fish meat used in our formulations and supports its role as an effective source of high-quality nutrients in the fortified breadsticks (3. Results and discussion).
C7: The manuscript states that sensory evaluation was performed by ten trained panelists; however, it does not describe how inter-panelist reliability or agreement was ensured. To validate the sensory results, the authors should clarify the procedures used to ensure consistency and reproducibility among evaluators. This may include details such as: Calculation of inter-rater reliability metrics (e.g., Cronbach’s alpha, Kendall’s coefficient of concordance, or analysis of variance among raters)
R7: Thank you for this comment. In the revised manuscript, the sensory methodology has been clarified to indicate that inter-panelist agreement was evaluated using Kendall’s tau correlation. Kendall’s tau was calculated for all descriptive sensory attributes using Statistica software, and in every case τ = 1.00, indicating perfect agreement among assessors. This confirms that the panelists ranked the samples consistently and that the sensory results are highly reliable. Appropriate details have been added to the Methods section (3.10. Sensory evaluation; L241-248; 3.11. Statistical analysis; L251-256).
C8: Justify the use of parametric tests (ANOVA), was the normality distribution checked before applying this statistical model
R8: Thank you for this important comment. The assumptions for applying parametric tests were checked before conducting the ANOVA. Data were evaluated for normality. Variables that met these assumptions were analyzed using one-way ANOVA with Tukey’s post-hoc test.
C9: Table 2 presents the proximate composition of the breadsticks, including fat, protein, ash, and fiber; however, the nitrogen-free extract (NFE), which represents the carbohydrate fraction of the product — is missing. Since NFE is a key nutritional parameter in proximate analysis and contributes significantly to the energy value and overall macronutrient balance of the snack, it should be included for completeness.
R9: Thank you for this important comment. In the revised manuscript, NFE has been calculated using the standard difference method [NFE% = DM% - (LC%+PC%+AC%+FC%)] and added to Table 2. Including NFE improves the completeness of the nutritional profile and ensures a more accurate representation of the macronutrient composition of the breadsticks (2.2. Basic composition; L129-140; 3.1. Basic composition L273-277; Table 2).
C10: In line 241, the authors refer to phosphorus and calcium as macronutrients; however, this terminology is inaccurate. These elements should be correctly referred to as macrominerals
R10: Thank you for this correction. We agree with the Reviewer’s observation. In line 286, the terms “macronutrients” for phosphorus and calcium have been replaced with the correct terminology, “macrominerals.” The revised manuscript now reflects the accurate classification of these elements.
C11: The results section would benefit from including the proportionate changes in macro- and micronutrient composition between the control and fortified breadstick samples. Presenting these changes, authors can add percentage increases or decreases relative to the control.
R11: Thank you for this helpful suggestion. In the revised manuscript, we have added proportional (percentage) changes relative to the control sample to highlight the magnitude of increases or decreases in macro- and micromineral contents. These percentage differences have been incorporated directly into the Results narrative to improve clarity and allow readers to better appreciate the nutritional impact of carp fortification (3.1. Basic composition; L291-313).
C12: The authors report an increase in AV after fish remains addition; however, the manuscript should also address potential side effects, such as increased lipid peroxidation and reduced shelf life
R12: Thank you for this valuable observation. The text discussing acid value has been expanded to address the potential implications of increased AV, including susceptibility to lipid peroxidation and possible effects on shelf life. An additional reference has been incorporated to support these points (3.3. Fatty acids profile and lipid values; L397-400).
C13: The authors claim that the addition of fish proteins produces antioxidant peptides, yet simply incorporating fish remains without prior enzymatic hydrolysis is unlikely to generate these bioactive peptides. The manuscript lacks any evidence from proteolytic treatment or controlled hydrolysis experiments that could support this mechanism. Furthermore, the reported antioxidant effect has not been validated in a biological system, and there is no data on the bioavailability or in vivo release of these peptides. Without such validation, it is unclear whether the product would exert the claimed antioxidant effects in the body. The discussion should also address potential limitations, including variability in peptide release and stability, as well as the possible impact on shelf life and lipid peroxidation
R13: We thank the Reviewer for this important observation. We agree that incorporating carp meat without applying controlled enzymatic hydrolysis does not guarantee the formation of bioactive antioxidant peptides, and that our results reflect in vitro antioxidant capacity only. In the revised manuscript, we have clarified this limitation (3.2. Antioxidant activity; L326-354).
C14: Table 6 is not completely visible (should be readjusted)
R14: Thank you for pointing this out. Table 6 has been reformatted and adjusted to ensure full visibility and proper alignment in the revised manuscript (Table 6).
C15: The amino acid analysis could be strengthened by comparing the results with the nutritional requirements of the target population and FAO standards, which would enhance the nutritional relevance and value of the study.
R15: Thank you for this constructive suggestion. The amino acid discussion has been expanded to compare the amino acid composition of the fortified breadsticks with FAO reference patterns and nutritional requirements of the target population. This comparison strengthens the nutritional relevance of the findings and demonstrates how carp enrichment enhances the protein quality of cereal-based snacks (3.4. Amino acid profile and loss of available lysine; L483-500).
Reviewer 2 Report
Comments and Suggestions for Authors
The manuscript "Nutritional Composition, Bioactive Properties, and Sensory Evaluation of Breadsticks Enriched with Carp Meat (Cyprinus carpio, L)" is an interesting topic for food science.
There are a few suggestions for revision:
- The abstract should be formulated without divisions such as ”objectives; results; conclusions”.
- The introduction is rather brief; it would be advisable to include information about other food fortified with Cyprinus carpio and what the effect were. Also, in the introduction section, it is advisable to follow the journal template, spacing, and so on.
- How were the percentages of fish used in the formulation chosen?
- I table 1, I notice that there are differences between samples for some ingredients, such as flour and water. From what I understand, you are only following the influence of different percentages of fish in the product. What could be reason for these differences between samples such as flour and water?
- Under the tables, it would be helpful to include the names of the samples (what do S5, S10 etc)
- How were the averages compared? Column or row?
- The discussion of the results is rather brief. Please insist on improving this aspect.
- The conclusion section needs to reworded to be more convincing and to include more accurate data from the results.
The article is well thought out, but need major revision. Please the follow the guidelines provided in the template offered by the Foods journal. For the Introduction section, please take into account the guidelines provided above, as well as for the other sections. Please clearly define the purpose of the study in accordance with the results. Kind regards!
Author Response
Response to the Reviewers' comments on manuscript ID: foods-4009058
Authors would like to thank for the comments to our journal submission entitled: „Nutritional Composition, Bioactive Properties, and Sensory Evaluation of Breadsticks Enriched with Carp Meat (Cyprinus carpio, L.)”. We appreciate the astute observations and constructive comments which helped us to improve our submission.
Changes made to the manuscript are marked in red, and the response includes detailed locations of the changes.
Reviewer #2
Comment 1: The abstract should be formulated without divisions such as ”objectives; results; conclusions”.
Response 1: We appreciate the reviewer’s suggestion. In accordance with the journal’s guidelines, the abstract has been reformulated into a single, continuous paragraph without subheadings. The revised version now presents the study’s purpose, key findings, and main conclusions in an integrated and cohesive format (Abstract).
C2: The introduction is rather brief; it would be advisable to include information about other food fortified with Cyprinus carpio and what the effect were. Also, in the introduction section, it is advisable to follow the journal template, spacing, and so on.
R2: We appreciate the Reviewer’s suggestions. The Introduction has been extended to include additional literature on previous food products fortified with common carp. Furthermore, the entire Introduction section has been checked according to the MDPI FOODS template, ensuring correct spacing, paragraph structure, and formatting requirements as outlined in the journal’s Instructions for Authors (1. Introduction).
C3: How were the percentages of fish used in the formulation chosen?
R3: Thank you for raising this important point. The selected inclusion levels of carp meat (0–30%) were determined through preliminary experiments in which several higher and lower percentages were tested. Additions above 30% resulted in excessively sticky dough, and poor structural integrity after baking, making the product technologically unsuitable. Levels below 5% produced minimal nutritional enhancement and only slight changes in protein quality.
C4: I table 1, I notice that there are differences between samples for some ingredients, such as flour and water. From what I understand, you are only following the influence of different percentages of fish in the product. What could be reason for these differences between samples such as flour and water?
R4: Thank you for this insightful observation. The differences in flour and water quantities among formulations are intentional and result from the need to maintain consistent dough yield, texture, and processing characteristics as the proportion of carp meat increased. Carp meat contains intrinsic moisture and has a different protein and fat composition compared with wheat flour; therefore, increasing the amount of fish necessarily alters the hydration properties and the binding capacity of the dough. To ensure comparable dough consistency, machinability, and final product weight across all treatments, the amount of water was gradually reduced and the flour content slightly adjusted as fish addition increased.
Such adjustments are standard in studies incorporating animal proteins or fish materials into cereal matrices, as the introduction of high-moisture, high-protein ingredients affects gluten development and dough rheology. To prevent excessive stickiness, maintain the intended texture, and allow proper shaping and baking, it was necessary to modify the supporting ingredients proportionally.
C5: Under the tables, it would be helpful to include the names of the samples (what do S5, S10 etc)
R5: Thank you for this suggestion. To improve clarity and readability, we have added a brief explanation of the sample codes directly below the tables (e.g., S0 - Breadsticks without carp meat, etc.). This clarification has been consistently included in all tables of the revised manuscript (Table 1).
C6: How were the averages compared? Column or row?
R6: Thank you for your comment. All comparisons in the tables were made within each row, meaning that the different samples (S0–S30) were compared for the same parameter. This has now been stated in the table footnotes in the revised manuscript (All tables).
C7: The discussion of the results is rather brief. Please insist on improving this aspect.
R7: Thank you for this comment. The Discussion section has been substantially expanded to provide deeper interpretation of the findings, clearer connections to published literature, and broader nutritional and technological context. The revised Discussion now more thoroughly explains the mechanisms underlying the observed changes in composition, antioxidant activity, lipid stability, amino acid and lysine retention, fatty acid distribution, mineral content, and sensory perception (3. Results and discussion).
C8: The conclusion section needs to reworded to be more convincing and to include more accurate data from the results.
R8: Thank you for this constructive recommendation. The Conclusion section has been rewritten to provide a more data-driven synthesis of the study’s findings (4. Conclusion).
C9: The article is well thought out, but need major revision. Please the follow the guidelines provided in the template offered by the Foods journal. For the Introduction section, please take into account the guidelines provided above, as well as for the other sections. Please clearly define the purpose of the study in accordance with the results. Kind regards!
R9: We sincerely thank the Reviewer for the positive assessment and constructive recommendations. In response, the manuscript has been thoroughly revised to follow the FOODS journal template and formatting guidelines. The Introduction has been expanded and reorganized according to the Reviewer’s suggestions, including a clearer scientific context, additional literature support, and improved transitions. All other sections (Methods, Results, Discussion, Conclusions) have likewise been reviewed and adjusted to match the journal’s structure, style, and formatting requirements. We appreciate the Reviewer’s guidance, which has substantially improved the quality and readability of the manuscript.
Round 2
Reviewer 1 Report
Comments and Suggestions for Authors
The suggested changes have been made and the manuscript has significantly improved after these changes.
Reviewer 2 Report
Comments and Suggestions for Authors
The authors followed the guidelines and instructions provided by the reviewer.